# External Validation of Mortality Scores among High-Risk COVID-19 Patients: A Romanian Retrospective Study in the First Pandemic Year

**DOI:** 10.3390/jcm11195630

**Published:** 2022-09-24

**Authors:** Amanda Rădulescu, Mihaela Lupse, Alexandru Istrate, Mihai Calin, Adriana Topan, Nicholas Florin Kormos, Raul Vlad Macicasan, Violeta Briciu

**Affiliations:** 1Department of Epidemiology, “Iuliu Haţieganu” University of Medicine and Pharmacy, 400348 Cluj-Napoca, Romania; 2The Teaching Hospital for Infectious Diseases, 400348 Cluj-Napoca, Romania; 3Department of Infectious Diseases, “Iuliu Haţieganu” University of Medicine and Pharmacy, 400348 Cluj-Napoca, Romania

**Keywords:** COVID-19, mortality scores, validation, Romania, comorbidities

## Abstract

Background: We aimed to externally validate three prognostic scores for COVID-19: the 4C Mortality Score (4CM Score), the COVID-GRAM Critical Illness Risk Score (COVID-GRAM), and COVIDAnalytics. Methods: We evaluated the scores in a retrospective study on adult patients hospitalized with severe/critical COVID-19 (1 March 2020–1 March 2021), in the Teaching Hospital of Infectious Diseases, Cluj-Napoca, Romania. We assessed all the deceased patients matched with two survivors by age, gender, and at least two comorbidities. The areas under the receiver-operating characteristic curves (AUROCs) were computed for in-hospital mortality. Results: Among 780 severe/critical COVID-19 patients, 178 (22.8%) died. We included 474 patients according to the case definition (158 deceased/316 survivors). The median age was 75 years; diabetes mellitus, malignancies, chronic pulmonary diseases, and chronic kidney and moderate/severe liver diseases were associated with higher risks of death. According to the predefined 4CM Score, the mortality rates were 0% (low), 13% (intermediate), 27% (high), and 61% (very high). The AUROC for the 4CM Score was 0.72 (95% CI: 0.67–0.77) for in-hospital mortality, close to COVID-GRAM, with slightly greater discriminatory ability for COVIDAnalytics: 0.76 (95% CI: 0.71–0.80). Conclusion: All the prognostic scores showed close values compared to their validation cohorts, were fairly accurate in predicting mortality, and can be used to prioritize care and resources.

## 1. Introduction

The COVID-19 pandemic brought an unprecedented high pressure on the medical systems; thus, an effective triage of patients regarding the risk of progressive deterioration is compelling for clinical decision making and effective resource allocation, including that for hospital beds, critical care resources, and targeted drug therapies.

Older adults and patients with previous comorbid conditions suffered greatly from SARS-CoV-2 infections, many of them being admitted to the intensive care unit (ICU), with in-hospital or ICU mortality up to 26–28% [1,2,3]. In very old patients, the survival probability at 30 days was found to be low when compared with that of ICU non-COVID-19 patients: 38% (35–42%) versus 57% (55–60%) [4]. Prognostic scores have been developed and tested in different populations since the beginning of the COVID-19 pandemic for better ICU resource management and to support clinicians in the discussion about prognosis with the patient’s family. A systematic review released in July 2020 found many published prognostic scores estimating the mortality risk in COVID-19 patients, with a high or unclear risk of bias, of which the 4CM Score was considered promising [5]. Other scores were recently proposed [6,7,8,9]. The worldwide applicability of these predictive scores remains an open question because healthcare systems and patient profiles differ between countries, and may impact the scores’ performance [10,11,12,13,14,15].

The aim of this study was to evaluate the validity and death risk stratification of selected prognostic scores in Romanian adult patients with significant comorbidities and severe-to-critical COVID-19.

## 2. Methods

### 2.1. Study Design and Setting

We conducted a retrospective study using the records retrieved from the Teaching Hospital of Infectious Diseases, Cluj-Napoca, Romania, and the main referral center for COVID-19 in Cluj County, Romania.

We included adult patients admitted in our hospital with a positive SARS-CoV-2 polymerase chain reaction (RT-PCR) nasal swab test, during the first pandemic year (1 March 2020–1 March 2021). A total of 4813 asymptomatic, mild, moderate, severe, and critical SARS-CoV-2-infected patients were hospitalized during the first pandemic year. We selected all the deceased patients from the 2937 patients with severe and critical COVID-19, classified according to the WHO Clinical management of severe acute respiratory infection (SARI) when COVID-19 disease is suspected [16] and at least two comorbidities. We included patients with two comorbidities because, according to the mortality score developed and validated by Knight et al., they have a higher risk of death [17]. For each of them, we retrieved from the hospital electronic database the first two controls matched by age at index date, sex, month of hospitalization, and at least two comorbidities. The comorbidities were defined by the Charlson Comorbidity index, with the addition of clinician-defined obesity [17,18]. There was a slightly higher number of women (17), to ensure exact matches for age and month of hospitalization.

### 2.2. Data Management and Study Outcome

We evaluated the predictive ability of three COVID-19 scores for in-hospital mortality in a retrospective study on COVID-19 patients. We had chosen these scores based on data availability, performance, and validation in previous studies [10,11,12,13,14,17,19,20,21,22,23].

The first prognostic score was the 4C Mortality Score (4CM Score) with reference to the International Severe Acute Respiratory and Emerging Infection Consortium [24] because it showed high discrimination with an area under the receiver-operating characteristic curve (AUROC) in the validation cohort of 0.79 (95% confidence interval: 0.77, 0.76 to 0.77). This score is the sum of the points assigned to the age, sex, number of comorbidities, respiratory rate, peripheral oxygen saturation level, Glasgow Coma Scale score, blood urea nitrogen level (BUN), and C-reactive protein (CRP) level. In the original 4CM Score, 2 points were given if the peripheral oxygen saturation in room air was below 92%, but not all the patients had saturation data in room air. Thus, we considered patients with <92% saturation, regardless of oxygen therapy, to have a score of 2 points. The 4CM Score ranges from 0 to 21, with the risk groups defined as low (0–3), intermediate (4–8), high (9–14), and very high (≥15) [17]. The 4CM Score was considered for further analysis.

The second prognostic score, the COVID-GRAM Critical Illness Risk Score (C-GRAM Score), was chosen because of its good accuracy, AUROC (95% CI: 0.880 (0.840–0.930)), and translation into an online risk calculator that was freely available [25], with criteria including age, X-ray abnormality, hemoptysis, dyspnea, unconsciousness, number of comorbidities, cancer history, neutrophil-to-lymphocyte ratio (NLR), lactate dehydrogenase (LDH), and direct bilirubin [21,26,27].

The third prognostic score, the COVID-19 Mortality Risk Score COVIDAnalytics (COVIDAnalytics), was validated in the US and Europe and translated into an online risk calculator freely available [28]. The score demonstrated good accuracy (AUROC: 0.81–0.92) and included many variables: age, sex, comorbidities, temperature, oxygen saturation, aminotransferases, creatinine, sodium, potassium, blood glucose levels, BUN, hemoglobin, leukocytes, platelet count, CRP, and prothrombin time [22].

We used demographic data, clinical data, and laboratory data included in the above-mentioned scores at presentation or on the first day of in-patient records. The COVID-19 severity was based on the discharge diagnosis, while all the clinical and laboratory parameters recorded at admission were used in the score’s calculation. We set the end point on the inclusion window to in-hospital death or discharge at home with improved health status.

### 2.3. Statistical Analysis

Descriptive statistics are presented as the medians (with interquartile ranges) or as absolute numbers (percentages) as appropriate. Univariable analysis performed with the Mann–Whitney test and odds ratios with 95% confidence intervals and Fisher’s test were used as applicable to compare the scores, demographics, comorbidities, and clinical and laboratory data between deceased patients and survivors. We calculated the scores for all the patients. For each score, receiver-operating curves (ROCs) were used to establish the cut-offs for which we calculated the sensitivity, specificity, and overall accuracy in predicting in-hospital mortality. Youden’s index was used to estimate the test performance with the optimal cut-off points and the corresponding sensitivity, specificity, and positive and negative predictive values at selected threshold values. The actual (observed) and generalized linear model (GLM) predicted death rates at each possible score value were computed for the selected scores. All the statistical analyses and plots were performed in the R 4.1.1 software environment for statistical computing and graphics. *p*-values were considered statistically significant when under 0.05 [29].

## 3. Results

During the study period, the in-hospital mortality rate was 3.7% (178 deceased/4813 asymptomatic, mild, moderate, severe, and critical cases) and 22.8% for severe-to-critical cases (178/780). Among the deceased patients, 158 fulfilled the severity criteria and had at least two comorbidities. The dataset included 474 patients, aged between 33 and 99 years, of which 408 (86%) patients were over the age of 60. A statistically significant difference between physiological parameters at admission and the laboratory results of selected tests in the non-survivor versus survivor subgroup was described. Malignancies, diabetes mellitus, chronic moderate or severe liver or renal disease, and chronic pulmonary disease were more frequently encountered in non-survivors, while obesity, hypertension, and chronic cardiac diseases were similar in both groups (Table 1).

The majority of our patients, 87% (412/474), were classified according to the 4CM Score at the high and very high risk levels (Table 2).

## 4. AUROC Analyses

All the scores used similar prognostic parameters, showing performance metrics (area under the receiver-operating characteristic curve—AUROC) close to 0.75 (Table 3, Figure 1 and Figure 2).

The 4CM Score showed the maximal accuracy of 0.72 (95% CI: 0.67–0.77%) at the cut-off of 14, with a sensitivity (Se) of 42.4%, specificity (Sp) of 86.7%, positive predictive value (PPV) of 61.5%, and negative predictive value (NPV) of 75.1%. Using the Youden’s index as the measure of test performance, the 13-point threshold was optimal, with the following parameters: accuracy: 0.71; Youden’s index: 34.2%; Se: 55.1%; Sp: 79.1%; PPV: 56.9%; NPV: 77.9% (Appendix A).

COVID-GRAM showed a maximal accuracy of 0.75 (95% CI: 0.69–0.79) at the cut-off of 90, with Se: 43.0%; Sp: 90.5%; PPV: 69.4%; and NPV: 76.1%. Using the Youden’s index as the measure of test performance, the 70-point threshold was optimal, with the following parameters: accuracy: 0.68; Youden’s index: 38.0%; Se: 71.5%; Sp: 66.5%; PPV: 51.6%; NPV: 82.4% (Appendix A).

COVIDAnalytics performed the best of all, with an AUROC of 0.76 (0.71–0.80) at the cut-off of 50 points, with Se: 44.7%; Sp: 88.9%; PPV: 66.0%; and NPV: 77.0%. Using the Youden’s index as the measure of test performance, the 35-point threshold was optimal, with the following parameters: accuracy: 0.692; Youden’s index: 40.8%; Se: 73.7%; Sp: 67.1%; PPV: 51.9%; NPV: 84.1% (Appendix A).

In the logistic model, the death rates and distribution of cases/non-survivors according to the 4CM Score are presented in Figure 3.

## 5. Discussion

In Romania, according to the published data, from the beginning of the pandemic until the 1 March 2021, there were 804,090 reported COVID-19 cases and 20,403 deaths, with a fatality rate of 2.54%. As of 1 March 2021, in other European countries, e.g., the United Kingdom, 4,190,000 cases and 123,039 deaths occurred, with a case fatality rate of 2.94%, and in Germany, there were 2,450,000 cases and 70,105 deaths, with a fatality rate of 2.87%, while the testing rates were 1.5/1000, 9.4/1000, and 2/1000, respectively. At the same time point, the mortality rates were 1066/million in Romania, 1803/million in the UK, and 835/million in Germany. Until the beginning of March 2021, Romania showed a similar fatality rate compared with other European countries, in the context of the lowest testing rate and a cumulative mortality rate lower than that in the UK, and higher than that in Germany and the average rate across European Union countries [30,31].

During the first pandemic year, the differences in the SARS-CoV-2 variants of concern (VOCs) might have influenced patients’ prognosis. From the third week of 2021, the Alpha VOC represented 20% of the sequenced SARS-CoV-2 samples in Romania, reaching 80% in week 8, the end of our study period. It is highly probable that, in the first and second waves, the wild variant was predominant, while in the third one, the Alpha/B.1.1.7 variant was prevailing, as other VOCs were very rarely identified in Romania [32,33]. Frampton et al. found no difference in disease severity between the ancestral virus and B.1.1.7 alpha lineage in London, and we presume that, in our study group and during the first pandemic year, the different variants did not influence the prognosis [34].

In a systematic review performed by Wynants et al. on 107 prognostic models for predicting mortality risk, progression to severe disease, intensive care unit admission, ventilation, intubation, or length of hospital stay, the 4CM Score was found to be the most promising model [5,17]. The scoring system for the prediction of in-hospital mortality was developed based on data obtained from large derivation and validation cohorts across 260 hospitals from England, Scotland, and Wales showing consistent discrimination, calibration, and clinical utility [17,20]. Compared with the original study, we found an AUROC of 0.72 (0.67–0.77), close to the original score confirmed in the validation cohort of 22,361 patients (AUROC: 0.77 (0.76–0.77)) [17]. Moreover, we used the 4CM Score to evaluate the probability of in-hospital death for each risk level. In our patients, those at the highest risk level, of at least 15, had 61% mortality, the same value as that found in the original study, and also a similar positive predictive value of 68% versus 62% [17]. Adderley et al. aimed to develop and externally validate novel prognostic models for adverse outcomes in the UK and externally validated the existing 4CM Score, adding more comorbidities, and the conclusion was that the new model’s performance was not significantly better when compared with the original score. They found an AUROC slightly higher than the value obtained in our study (0.753 (95% CI: 0.720 to 0.785)) [6].

We had chosen the other scores based on data availability, performance, and validation in previous studies [14,21,22]. Among 14,343 French patients, Lombardi et al. evaluated 32 prognostic scores, and found that seven prognostic scores were fairly accurate in predicting death in hospitalized COVID-19 patients, including the 4CM Score and OVID-GRAM. The 4CM Score stood out, as it performed as well as in the initial validation cohort, during the first epidemic wave and subsequent waves, and in younger and older patients, with an AUROC of 0.785, while the COVID-GRAM showed an AUROC of 0.700 [14]. In our study, we found a lower value of the AUROC for the 4CM Score and a better performance for COVID-GRAM (AUROC: 0.74 (0.69–0.79)).

In older patients, Covino et al. found higher AUROCs for the 4CM Score and COVID-GRAM (0.799 (0.738–0.851)) and (0.785 (0.723–0.838)), respectively [10]. Van Dam et al. evaluated, in a secondary/tertiary medical center from the Netherlands, eleven prediction models and found that the 4CM Score showed very good discriminatory performance for 30-day mortality (AUROC: 0.84 (0.79–0.88)) [11]. In Japanese patients with pre-existing cardiovascular diseases, Kuroda et al. found a very good performance for the 4CM Score (AUROC: 0.84 (95% CI 0.80–0.88)) [13]. Jones et al. found, in hospitalized Canadian patients, a higher 4CM Score, with an AUROC of 0.77 (95% CI: 0.79–0.87), and at the cut-off of 14, the test accuracy and predictive values were similar to ours [12]. The overall mortality rates across the risk groups were identical to ours for the low (0%) and high risk levels (27.2% versus 27%) but lower for the intermediate risk (8% versus 13%) and very high risk (54.2% versus 61%) [12].

Using the same score in Saudi Arabia, Mohamed et al. found an AUROC of 0.9 (95% CI: 0.859 to 0.954), 71% sensitivity, and a specificity of 88.6% but an underestimated mortality rate among the very high risk level (66.2% versus 90%) [15].

Compared with the previous scores, we found a slightly greater discriminatory ability for COVIDAnalytics, with an AUROC of 0.76 (95% CI: 0.71 to 0.80), close to the value found in the United States (AUROC: 0.81 (95% CI: 0.76–0.85)) but lower compared with the values in the validation cohorts from Spain and Greece [22].

The external validation of the 4CM Score in different populations might bring inference, and our results are in line with the previous validation studies favoring the conclusion that the model can be used regardless of ethnicity and healthcare systems. In the original study, the 9-point cut-off was considered to rule out or rule in mortality, respectively, with an NPV of ~90% and PPV of ~40% [17]. Since our patients were older (65.8% > 70 years) compared with the initial derivation cohort (57.3%, *p* = 0.0002), the cut-off value for the best performance was found to be 13 points (Sn: 55%; Sp: 79%; PPV: 57%; NPV: 77.9%), similar to the values obtained in the original study at the same threshold (Sn: 62.5; Sp: 75.3%; PPV: 52%; NPV: 82.3%) [17].

The evaluated scores (4CM Score, COVID-GRAM, and COVIDAnalytics) included laboratory tests, of which CRP, LDH, and kidney damage (BUN and creatinine level) are well-known predictors for severe disease and were also confirmed in our evaluation, but the usefulness of the liver damage assessment seems to be irrelevant [35,36,37].

In our selected population of patients above 60 years (88%), the weight of age in the scores’ calculation was probably reduced because of the limited and advanced age range. This may partially explain why the PPVs for the in-hospital mortality of all the scores were lower than those in the original reports and other validation studies [10,11,13,14,17,21].

All the validated scores (4CM Score, COVID-GRAM, and COVIDAnalytics) showed a high NPV (77.9–84%), which may be used to exclude the risk of subsequent deterioration in patients designated to a non-critical area. We emphasize the 4CM Score as the most valuable because of its good performance, ease of use, and death risk stratification. Additionally, we obtained similar values per risk level, mainly for the high and very high levels, to in the original study [17]. The other prognostic scores showed similar performance; still, the 4CM Score allowed risk ranking for in-hospital death, which added a final evaluation beyond the negative and positive predictive values at the best cut-off values.

Our data confirmed that all the patients showed an overall high comorbidity index, with chronic pulmonary, renal, and liver diseases, cancers, and diabetes mellitus being associated with a higher risk for poor outcomes. Common conditions present in older patients such as hypertension, obesity, and chronic cardiac disease were not found to be death predictors, partially because these conditions were encountered in many of our patients.

The first study that investigated the patterns of comorbidities for SARS-CoV-2 fatalities in Romania showed that male gender, hypertension, diabetes, obesity, and chronic kidney disease were most frequently associated with COVID-19 fatalities [38].

According to the Report upon the State of Health in the EU 2021, in Romania, in 2018, more than a third of deaths were cardiovascular, and in 2021, ischemic heart disease (19.1%) and stroke (16.3%) were the leading causes of death. Life expectancy increased by more than four years between 2000 and 2019 but declined during the pandemic by 1.4 years in 2020 due to COVID-19 [30].

In our study group, the median comorbidity value was 5, with diabetes mellitus and cancer associated with a higher risk of death (OR: 1.62 [1.09, 2.39] (*p* = 0.020) and 2.63 [1.45, 4.75] (*p* = 0.001), respectively). In Romania, the prevalence of diabetes is approximately 11.6%, and that of prediabetes is double [39]. Several studies and meta-analyses have investigated the impact of diabetes mellitus on the severity of COVID-19 [40,41,42]. Although diabetes is an independent risk factor for disease severity, patients with diabetes often have other comorbidities or concurrent factors that could add even more risk for severe COVID-19 [43,44,45].

Regarding malignancies, many studies demonstrated that the risk of death related to COVID-19 was not increased if the survival was longer than 5 years, but an increased risk of death was found for recent cancers (<1 year) and hematological malignancies, with an aHR of 1.72 (95% CI: 1.5–1.96) and aHR of 2.8 (95% CI: 1.5–1.96), respectively [46]. Docherty et al., within the ISARIC WHO Clinical Characterization Protocol conducted with 20,133 patients, found an aHR for malignancies of 1.13 (95% CI: 1.02–1.24), and also, Orlando et al., in a large Italian study, found a similar value, with an OR of 1.18 (1.01–1.37) [19,47]. In our study, an increased risk of death in cancer patients was observed (OR: 2.63; 95% CI [1.45, 4.75]), probably in the context of the increasing trend in the last decade in Romania, with 7.8% proportionate mortality for the main cancers (lung, breast, and colorectal) in 2021 [30].

In large studies, chronic cardiac disease, chronic non-asthmatic pulmonary disease, chronic kidney disease, obesity, chronic neurological disorders (such as stroke), dementia, and liver disease were associated with increased in-hospital mortality [19,46,48].

We also found that chronic non-asthmatic pulmonary disease, chronic kidney disease, and moderate or severe liver disease were significantly more frequently encountered in the deceased patients than in the survivors (ORs of 6.59, 2.95, and 2.26, respectively).

Somewhat surprisingly, our results did not show an association between obesity and in-hospital mortality due to the high overall rate, but, at the national level, by 20 February 2021, obesity was present in 18.7% of all the COVID-19-related deaths [49]. Additionally, hypertension and chronic cardiac disease were present in both study groups at very high rates; hence, no significant statistical difference was found. According to studies published by Williamson et al. and Docherty et al. on comorbidities and the risk of death, hypertension was not found to be a risk factor, but chronic heart diseases were associated with a higher risk of death (HR: 1.17 (1.12–1.22) and HR: 1.16 (1.08–1.24), respectively) [19,48].

Although all the prognostic scores showed close values compared to their derivation and validation cohorts and were fairly accurate in predicting mortality in hospitalized Romanian COVID-19 patients in the first year of the pandemic, the validation of novel clinical risk prediction models to estimate the risk of COVID-19-related mortality in vaccinated people or previously infected people is essential in the present phase of the pandemic, when a large proportion of the population at risk is vaccinated, even with booster doses, and reinfections with new VOCs are rather frequent.

### Study Limitations and Strengths

The retrospective nature of the study, even with its good inclusion/exclusion criteria, made it prone to bias. Secondly, the sample was not large enough and included mainly older patients; therefore, the inference for the general population might be less relevant, even though the patients were selected from a large tertiary center in NW Romania, and also, we did not evaluate more pandemic waves. However, our estimations were comparable to those in other validation studies from different populations and showed a good reliability.

## 6. Conclusions

All the evaluated scores calculated at admission demonstrated good performance in predicting in-hospital death in Romanian COVID-19 patients with comorbidities. The 4CM Score was the best model and can be easily integrated into electronic medical records systems to calculate each individual patient’s probability of death or ICU admission, and can be used to prioritize care and resources.

## Figures and Tables

**Figure 1 jcm-11-05630-f001:**
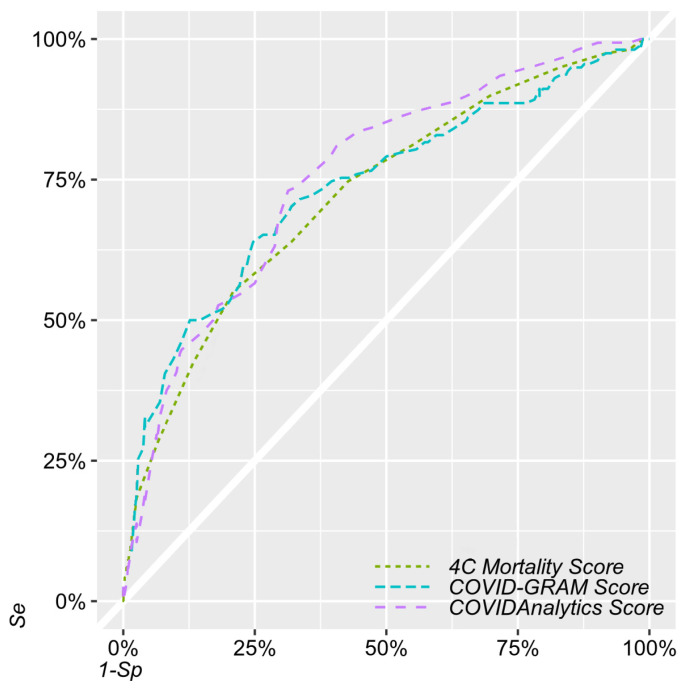
Receiver-operating curves (ROCs) comparing the original 4CM Score (dotted green line), COVID-GRAM (dashed blue), and COVIDAnalytics scores (dashed purple).

**Figure 2 jcm-11-05630-f002:**
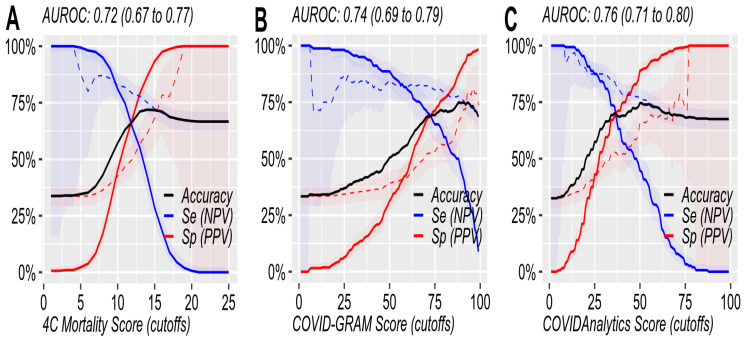
The parameters of the three mortality prediction scores for all the cut-offs. (**A**)—4CM Score, (**B**)—COVID-GRAM, and (**C**)—COVIDAnalytics. Black line: diagnostic accuracy; solid blue line: sensitivity; dashed blue line: negative predictive value; solid red line: specificity; dashed red line: positive predictive value.

**Figure 3 jcm-11-05630-f003:**
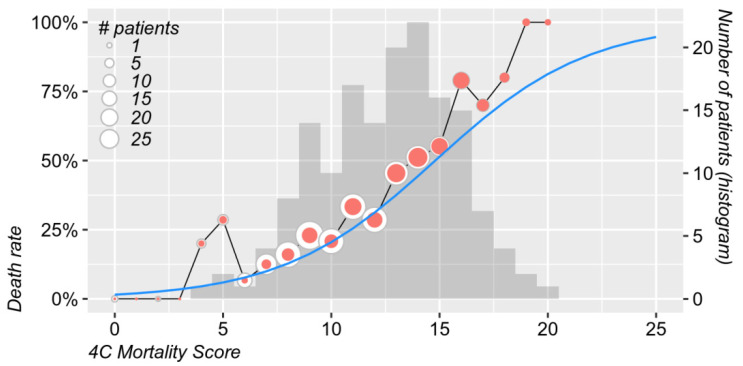
Death rate (left *y* axis) and distribution (right *y* axis) of the original 4CM Scores. Dot sizes are proportional to the number of patients with each score value, with the inner red dots showing the fatal cases. The smooth blue line shows the predicted death rate according to the logistic model (GLM).

**Table 1 jcm-11-05630-t001:** Summary statistics, by deceased vs. control group.

	Details	Deceased	Survivors	Total	Univariate Analysis *p* Value for the Applied Tests
		158	316	474	
**Age**	*n* (%) Median (IQR)	158 (100%) 75 (65–81)	316 (100%) 75 (65–81)	75 (65–81)	MW: *p* = 0.632
Gender female		67 (42.4%)	151 (47.8%)	218 (46.0%)	*p* = 0.266
**Risk scores**	Median (IQR)				
4CM Score		14 (11.25–16)	11 (9–13)	12 (10–14)	**MW: *p* < 0.001**
COVID-GRAM		87.32 (66.73–96.16)	61.94 (46.63–77.67)	67.73 (50.15–87.92)	**MW: *p* < 0.001**
COVIDAnalytics		48 (34–61.25)	28 (20–41.5)	34 (22.5–50)	**MW: *p* < 0.001**
**Physiological parameters at admission**					
Peripheral oxygen saturation	%	88.5 (80–94)	92 (88–96)	91 (85–95)	**MW: *p* < 0.001**
Respiratory rate (breaths/min)	Median (IQR)	29.5 (24–34)	27 (22–30)	28 (24–30)	**MW: *p* = 0.037**
Glasgow Coma score	Median (IQR)	7 (3–10.75)	13 (5.5–13.5)	8 (3–13)	**MW: *p* = 0.029**
Heart rate (beats per minute)	Median (IQR)	88.5 (75–100)	82 (74–95)	84 (74–96.25)	**MW: *p* = 0.042**
**Comorbidities**					
Number		5 (4–5)	5 (3–5)	5 (3–5)	**MW: *p* = 0.002**
Malignancy	*n* (%)	27 (17.1%)	23 (7.3%)	50 (10.5%)	**OR = 2.63 [1.45, 4.75] (*p* = 0.001)**
Diabetes mellitus	*n* (%)	71 (44.9%)	106 (33.5%)	177 (37.3%)	**OR = 1.62 [1.09, 2.39] (*p* = 0.020)**
Chronic pulmonary disease (not asthma)	*n* (%)	28 (17.9%)	10 (3.16%)	38 (8%)	**OR = 6.59 [3.21–13.38]** ** *p* ** **< 0.0001**
Chronic kidney disease	*n* (%)	32 (20%)	25 (7.9%)	57 (12%)	**OR = 2.95 [1.66–5.17]** ** *p* ** **= 0.0002**
Moderate or severe liver disease	*n* (%)	16 (10.1%)	15 (4.7%)	31 (6.5%)	**OR = 2.26 [1.10–4.7]** ** *p* ** **= 0.030**
Obesity and overweight	*n* (%)	55 (34.8%)	121 (38.3%)	176 (37.1%)	OR = 0.86 [0.57–1.29] *p* = 0.48
Hypertension	*n* (%)	120 (75.9%)	240 (75.9%)	360 (75.9%)	OR = 0.99 [0.63–1.54) *p* = 0.99
Chronic cardiac disease	*n* (%)	89 (56.3%)	153 (48.4%)	242 (51%)	OR =1.37 [0.93–2.03] *p*= 0.12
**Laboratory values**					
CRP (mg/L)	Median (IQR)	124.7 (56.5–210)	67.5 (23.1–142.9)	87.2 (34.5–161.9)	**MW: *p* < 0.001**
LDH (IU/L)	Median (IQR)	432 (327–569)	302.5 (226.25–402)	341 (241–466.5)	**MW: *p* < 0.001**
ALT (IU/L)	Median (IQR)	28 (19–60)	31 (19–48)	30 (19–49)	**MW: *p* = 0.918**
AST (IU/L)	Median (IQR)	41 (30–74)	33 (24–49)	36 (25–54)	**MW: *p* < 0.001**
BUN (mg/dL)	Median (IQR)	82 (57.1–131)	48 (37–71)	56.1 (39–89)	**MW: *p* < 0.001**
Creatinine (mg/dL)	Median (IQR)	1.35 (0.88–2.12)	0.9 (0.72–1.19)	0.98 (0.75–1.41)	**MW: *p* < 0.001**
Neutrophil count (×10^9^/L)	Median (IQR)	7.62 (4.58–11.75)	5.49 (3.34–8.95)	6.04 (3.59–9.71)	**MW: *p* < 0.001**
Lymphocyte count (×10^9^/L)	Median (IQR)	0.66 (0.44–1.02)	0.9 (0.61–1.28)	0.82 (0.54–1.21)	**MW: *p* < 0.001**

MW = Mann–Whitney test; OR = odds ratio [95% CI] and *p* value from Fisher test; for all numeric variables: median (interquartile range). ALT, alanine aminotransferase; AST, aspartate aminotransferase. CRP—upper normal value (UNV), 10 mg/L; BUN—UNV, 50 mg/dL; LDH—UNV, 250 IU/L; creatinine—UNV, 1.2 mg/dL; neutrophil count—range of values, 1.5–6.6 × 10^3^/µL; lymphocyte count—range of values, 1.1–3.5 × 10^3^/µL; ALT—UNV, 45 IU/L; ALT—UNV, 45 IU/L.

**Table 2 jcm-11-05630-t002:** Mortality rates according to the 4CMS risk levels.

4CM Score by Risk Level	Deceased 158	Survivors 316	Mortality per Risk Level		Mortality per Risk Level with the Original 4CM Score *
Low (0–3)	0	3 (0.9%)	0 (0%)	*p* = 0.22	1.2%
Intermediate (4–8)	8 (5%)	51 (16.1%)	8 (13%)	*p* = 0.0006	9.9%
High (9–14)	83 (52.5%)	220 (69.6%)	83 (27%)	*p* = 0.0003	31.4%
Very high (15–21)	67 (42.4%)	42 (13.3%)	67 (61%)	*p* < 0.0001	61.5%

* [17].

**Table 3 jcm-11-05630-t003:** Comparative performance of the three prognostic scores in our study and in their original cohorts.

	AUROC (95% CI)	AUROC in the Original Study
4C Mortality Score [17]	0.72 (0.67–0.77)	0.767 (0.760–0.773)
COVID-GRAM Score [21]	0.74 (0.69–0.79)	0.880 [0.840–0.930] *
COVIDAnalytics Score [22]	0.76 (0.71–0.8)	0.90 **

* COVID-GRAM Score AUROC was developed to predict the composite outcome of admission to the intensive care unit (ICU), invasive ventilation, or death. ** COVIDAnalytics score webpage did not provide any CI; a simplified version of the score was reported with an AUROC = 0.82.

## Data Availability

The data presented in this study are available on request from the principal author. The data are not publicly available due to privacy restrictions.

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
