# Peer review of "External Validation of Mortality Scores among High-Risk COVID-19 Patients: A Romanian Retrospective Study in the First Pandemic Year"

_jcm, 2022, doi:10.3390/jcm11195630_

Round 1

Reviewer 1 Report

Radulescu et al conducted a study assessing the external validation of a mortality score among high-risk COVID-19 patients. They found that the three prognostic scores (4CM, COVID-GRAM, COVIDAnalytics) were able to accurately predict mortality to help with resource prioritization.

This study is important but needs additional statistical descriptions and definitions for the reader to understand the take-home message.

Major revisions

·       A significant limitation is the outcome observed. For example, in-hospital mortality is not consistent, without a post-operative day cutoff. Additionally, the baseline populations of who had

·       Table 1 is confusing – please relabel the columns labeled ‘yes’ and ‘no’, and maintain appropriate capitalization

·       Are the odds ratios listed in Table 1 for multivariable or univariable analysis? Please include in the Methods section.

·       How were the variables in Table 1 defined? For example, how was ‘moderate or severe liver disease’ determined?

·       The first few paragraphs of the Discussion are more appropriate for the Introduction

·       The analysis needs to be more clearly described in order for the reader to follow along – for example, what is ‘accuracy’?

Minor revisions

·       In addition to describing resource prioritization, risk stratification helps us with informed consent to discuss prognosis with families.

·       The phrasing in the introduction is confusing – it would be helpful to stick with either mortality or survival rates, but not both. The sentence starting in line 42 is a run-on sentence.

·       What is the impact of resource prioritization? How do tools like risk prognostication help patients? The authors should describe these concepts in the introduction.

·       How were patients scored using 4CM when on supplemental oxygen with an O2 saturation >92%? How did the authors decide to delineate this undefined population?

·       Please keep significant figures consistent

·       Line 119 should be ‘statistically significant’

Author Response

Dear Editors and Reviewers:

Thank you very much for reviewing our manuscript numbered ID jcm-1877587 and we really appreciate your helpful comments and valuable suggestions, which not only helped us with the improvement of our manuscript, but also will facilitate our future researches. Based on these comments and suggestions, we carefully revised and improved our manuscript. We are now sending the new manuscript for your reconsideration to publish in JCM, and we hope the revisions will meet with approval. Our point-by-point responses to the comments and suggestions are as follows, and please see the new manuscript for details of revisions. All changes are highlighted in blue font in the revised manuscript.

Answers to Reviewer 1

Comments and Suggestions for Authors

Radulescu et al conducted a study assessing the external validation of a mortality score among high-risk COVID-19 patients. They found that the three prognostic scores (4CM, COVID-GRAM, COVIDAnalytics) were able to accurately predict mortality to help with resource prioritization.

This study is important but needs additional statistical descriptions and definitions for the reader to understand the take-home message.

Major revisions

1. A significant limitation is the outcome observed. For example, in-hospital mortality is not consistent, without a post-operative day cutoff. Additionally, the baseline populations of who had1.

R. All the study patients were evaluated at discharge and the outcome was death or discharged home improved. We could not evaluate the outcome at 28 or at 90 days after discharge. The aim of the study was to externally validate prognostic scores for COVID-19 during the acute phase of the disease.

2. Table 1 is confusing – please relabel the columns labeled ‘yes’ and ‘no’, and maintain appropriate capitalization. Are the odds ratios listed in Table 1 for multivariable or univariable analysis? Please include in the Methods section.

R: We have introduced the outcome: „Deceased” and „Survivors” (instead of „yes” and „no”) and Univariable analysis with P values for the applied tests (we did not performed multivariable analysis). Patients’ comorbidities are expressed as numbers and percentages and we calculated the odds ratio and p values from Fisher test. For the numeric variables we calculated the median (interquartile range) and we used Mann-Whitney test.

3. How were the variables in Table 1 defined? For example, how was ‘moderate or severe liver disease’ determined?

R: The comorbidities were defined according to the “Charlson comorbidity index”, and based on patient’s history and clinician’s diagnosis. E.g. Severe liver disease = cirrhosis and portal hypertension with variceal bleeding history, moderate = cirrhosis and portal hypertension but no variceal bleeding history.

4.The first few paragraphs of the Discussion are more appropriate for the Introduction

R: We modified as suggested, the first paragraph from Discussion was inserted in the Introduction section.

5. The analysis needs to be more clearly described in order for the reader to follow along – for example, what is ‘accuracy’?

R: We changed in Method as follows: „For each score, receiver-operating curves (ROC) were used to establish the cut-offs for which we calculated the sensitivity, specificity and overall accuracy in predicting in-hospital mortality.”

Minor revisions

1. In addition to describing resource prioritization, risk stratification helps us with informed consent to discuss prognosis with families.

R: We introduced this very good suggestion in „Introduction”: „Prognostic scores have been developed and tested in different populations since the beginning of the COVID-19 pandemic for a better ICU resource management and to support clinicians in the discussion about prognosis with patient’s family.”

2. The phrasing in the introduction is confusing – it would be helpful to stick with either mortality or survival rates, but not both. The sentence starting in line 42 is a run-on sentence.

R: The literature data was presented in both ways, either mortality or survival which was very modest in older patients. We presented the data as was mentioned by the authors in the reference papers.

We have modified the paragraph starting with line 42: „A systematic review released in July 2020 found many published prognostic scores estimating mortality risk in COVID-19 patients, with a high or unclear risk of bias, of which the 4CM score was considered promising [5]. Other scores were recently proposed [6-9].”

3. What is the impact of resource prioritization? How do tools like risk prognostication help patients? The authors should describe these concepts in the introduction.

R: Risk prognostication is usefull for the referral to ICU. During the Alpha and Delta waves we were confrunted with a shortage of ICU beds and risk stratification was useful as an anticipative tool for ICU beds in other hospitals. We already modified in Introduction. „Prognostic scores have been developed and tested in different populations since the beginning of the COVID-19 pandemic for a better ICU resource management and to support clinicians in the discussion about prognosis with patient’s family.”

4.How were patients scored using 4CM when on supplemental oxygen with an O2 saturation >92%? How did the authors decide to delineate this undefined population?

R: Any oxygen suplimentation needed is scored with 2 points according to 4CM score.

5. Please keep significant figures consistent

R: We could not find inconsistencies in our figures.

6. Line 119 should be ‘statistically significant’

R: We made the correction.

Thank you very much for your time and amendments.

Reviewer 2 Report

Amanda Rădulescu and colleagues conducted a retrospective clinical study to validate the risk prediction scores 4C Mortality Score (4CM Score), COVID-GRAM Critical Illness Risk Score (COVID-GRAM) and COVIDAnalytics in patients hospitalized with comorbidities and severe or critical COVID-19. While I find the study methodically sound and relevant, I would like to raise the following concerns:    

Minor concerns:

The study was conducted including only patients until March 2021. If possible, also more recent patients should be included to account for effects of vaccines and new SARS-CoV-2 variants, such as the omicron variant.

ll. 59-61: the authors state that they selected only patients with critical or severe COVID-19. It is not clear, at what point the severity was determined (e.g. on admission). They further state, that all 178 patients who died, were among the critical or severe patients (ll. 115-117). This suggests, that severity was determined not upon admission but at a later time point. It should be clarified, when the severity was determined and discussed why no patient died, who was hospitalized and diagnosed with asymptomatic or mild COVID-19.

The authors selected patients with at least two comorbidities for this study and severe or critical COVID-19. They should discuss and clarify the rationale behind this decision.

The authors should clarify how patients were selected as matched controls.

ll. 270-272: The authors state that “Our data confirmed that all patients showed an overall high comorbidity index, …”. Does this statement refer to all patients at the hospital or to the study cohort? If it refers specifically to the study cohort, this seems not remarkable at all, since patients with at least two comorbidities were selected. This should be clarified.

l. 294: the second CI seems to be incorrect and a duplicate of the first CI: the aHR is 2.8 but the 95% CI is 1.5-1.96. This should be corrected.

ll. 289-299: The authors relate the increased mortality of cancer patients in their study to “the increasing trend in the last decade”. It is unclear, to what trend the authors refer and how this impacts on this study in particular and not on other studies.

Author Response

Dear Editors and Reviewers:

Thank you very much for reviewing our manuscript numbered ID jcm-1877587 and we really appreciate your helpful comments and valuable suggestions, which not only helped us with the improvement of our manuscript, but also will facilitate our future researches. Based on these comments and suggestions, we carefully revised and improved our manuscript. We are now sending the new manuscript for your reconsideration to publish in JCM, and we hope the revisions will meet with approval. Our point-by-point responses to the comments and suggestions are as follows, and please see the new manuscript for details of revisions. All changes are highlighted in blue font in the revised manuscript.

Answers to Reviewer 2

Comments and Suggestions for Authors

Amanda Rădulescu and colleagues conducted a retrospective clinical study to validate the risk prediction scores 4C Mortality Score (4CM Score), COVID-GRAM Critical Illness Risk Score (COVID-GRAM) and COVIDAnalytics in patients hospitalized with comorbidities and severe or critical COVID-19.

While I find the study methodically sound and relevant, I would like to raise the following concerns:    

Minor concerns:

1. The study was conducted including only patients until March 2021. If possible, also more recent patients should be included to account for effects of vaccines and new SARS-CoV-2 variants, such as the omicron variant.

1. In Romania, until March 2021, the vaccination coverage was very low. Therefore, we did not introduce this item since all the patients with severe/critical disease were not vaccinated. Validation of novel clinical risk prediction models to estimate the risk of COVID-19 related mortality in vaccinated people or infected with new variants of concern will be performed in future studies as mentioned in discussion section (line 321-325).

2. 59-61: the authors state that they selected only patients with critical or severe COVID-19. It is not clear, at what point the severity was determined (e.g. on admission). They further state, that all 178 patients who died, were among the critical or severe patients (ll. 115-117). This suggests, that severity was determined not upon admission but at a later time point. It should be clarified, when the severity was determined and discussed why no patient died, who was hospitalized and diagnosed with asymptomatic or mild COVID-19.

2. We modified “The severity was based on discharge diagnosis, while all clinical and laboratory parameters recorded at admission were used in scores calculation.”. There were 178 patients who died from which 158 fulfilled the study inclusion criteria of at least two comorbidities.

3. The authors selected patients with at least two comorbidities for this study and severe or critical COVID-19. They should discuss and clarify the rationale behind this decision.

3. The decision to choose patients with at least two comorbidities was related to 4CM Score which was the first prognosis score considered. “We included patients with two comorbidities because, according to the mortality score developed and validated by Knight et al., they have a higher risk of death [17].”

4. The authors should clarify how patients were selected as matched controls.

4. “For each of them, we retrieved from the hospital electronic database the first two controls were matched by age at index date, sex, month of hospitalization, and at least two comorbidities” - included in Methods section.

5. 270-272: The authors state that “Our data confirmed that all patients showed an overall high comorbidity index, …”. Does this statement refer to all patients at the hospital or to the study cohort? If it refers specifically to the study cohort, this seems not remarkable at all, since patients with at least two comorbidities were selected. This should be clarified.

5. We presented in Table 1. the number of comorbidities in all patients with a Median of 5, the same for survivors and deceased, but the association of comorbidities might have had an impact on the outcome.

6. 294: the second CI seems to be incorrect and a duplicate of the first CI: the aHR is 2.8 but the 95% CI is 1.5-1.96. This should be corrected.

6. We presented two values, the first for recent cancers (< 1 year) and the second for haematological malignancies. „Regarding malignancies, many studies demonstrated that the risk of death related to COVID-19 was not increased if the survival was longer than 5 years, but increased risk of death was found for recent cancer (<1 year) and haematological malignancies aHR 1.72 (95%CI, 1.5-1.96) and aHR 2.8 (95% CI, 1.5-1.96), respectively [46]”.

7. 289-299: The authors relate the increased mortality of cancer patients in their study to “the increasing trend in the last decade”. It is unclear, to what trend the authors refer and how this impacts on this study in particular and not on other studies.

7. We wanted to emphasize the increasing trend of malignancies in Romania. We added this information in the text.

Thank you very much for the suggested references, but we tried to keep the articles related to the analysed scores.